# Do Different Patterns of Employment Quality Contribute to Gender Health Inequities in the U.S.? A Cross-Sectional Mediation Analysis

**DOI:** 10.3390/ijerph191811237

**Published:** 2022-09-07

**Authors:** Trevor Peckham, Noah Seixas, A. B. de Castro, Anjum Hajat

**Affiliations:** 1Hazardous Waste Management Program in King County, Seattle, WA 98104, USA; 2Department of Environmental and Occupational Health Sciences, University of Washington, Seattle, WA 98195, USA; 3School of Nursing, University of Washington, Seattle, WA 98195, USA; 4Department of Epidemiology, University of Washington, Seattle, WA 98195, USA

**Keywords:** gender inequities in health, employment quality, precarious employment, mediation analyses, latent class analysis

## Abstract

Compared to recent generations, workers today generally experience poorer quality employment across both contractual (e.g., wages, hours) and relational (e.g., participation in decision-making, power dynamics) dimensions within the worker–employer relationship. Recent research shows that women are more likely to experience poor-quality employment and that these conditions are associated with adverse health effects, suggesting employment relations may contribute to gender inequities in health. We analyzed data from the General Social Survey (2002–2018) to explore whether the multidimensional construct of employment quality (EQ) mediates the relationship between gender and health among a representative, cross-sectional sample of U.S. wage earners. Using a counterfactually-based causal mediation framework, we found that EQ plays a meaningful role in a gender–health relationship, and that if the distribution of EQ among women was equal to that observed in men, the probability of reporting poor self-reported health and frequent mental distress among women would be lower by 1.5% (95% Confidence Interval: 0.5–2.8%) and 2.6% (95% CI: 0.6–4.6%), respectively. Our use of a multidimensional, typological measure of EQ allowed our analysis to better account for substantial heterogeneity in the configuration of contemporary employment arrangements. Additionally, this study is one of the first mediation analyses with a nominal mediator within the epidemiologic literature. Our results highlight EQ as a potential target for intervention to reduce gender inequities in health.

## 1. Introduction

The changing structure and character of employment relations in industrialized economies have increasingly gained attention within public and occupational health research [1,2,3]. A primary concern relates to the widespread shift away from the so-called Standard Employment Relationship (SER; i.e., permanent, full-time, and adequately compensated employment) and toward employment practices that emphasize firms’ ability to exercise flexibility with respect to their labor force. Such practices often result in a diminishing of commitments and rewards—and an increased shouldering of risk—flowing from employers to workers. For many workers, this has meant decreased security and power across multiple dimensions of the employment relationship. Compared to the last several generations, workers today are more likely to experience non-permanent contractual arrangements; volatile and asocial work schedules; stagnant wages and decreased access to fringe benefits; less opportunity to develop socially valued skills; and generally more imbalanced employer–worker power dynamics, including the lack of collective bargaining or other involvement in workplace decision-making [1,4,5]. These labor market circumstances, which are proximally structured by the contractual and relational features of employment, are associated with a variety of adverse physical and mental health outcomes among working populations [1,6].

Related to these employment trends were significant changes in workforce demographics, notably the dramatic increase in the number of women engaged in paid labor [7]. However, women continue to face multiple layers of marginalization and disadvantage within the labor market, which are rooted in gendered inequities in power and persistent, stark divisions in paid and unpaid labor [8]. As such, women disproportionately experience poor-quality and precarious employment conditions [9,10,11]. In parallel, women have continually exhibited a gendered health–survival paradox, in which, compared to men, they systematically report higher levels of morbidities and worse health-related quality of life [12], but live longer [13]. In this study, we examine the quality of employment relations as a proximal mechanism of gender health inequities.

### 1.1. Employment Quality and Health

A significant development in public health research is growing attention to how the terms and conditions of worker–employer relationships affect health. It is useful to make a conceptual distinction between these employment conditions and more oft-studied working conditions [10,14,15]. Working conditions, which have been the focus of the vast majority of work and health research to date, describe the character of specific job tasks (e.g., whether work is monotonous) and the physical and psychosocial environmental settings in which work is performed (e.g., presence of chemical hazards, pressure to complete tasks rapidly). In contrast, employment conditions represent the formal and informal arrangements between workers and employers that determine both contractual (e.g., wages, hours) and relational (e.g., participation in decision-making, power dynamics) components of one’s job [16,17,18]. While early research on employment conditions and health has focused principally on the phenomena of employment instability—such as perceived fear of job loss or non-permanent contracts—an important advancement in this area is the understanding that multiple aspects of employment affect workers’ health [1,19].

In this study, we use the construct of employment quality (EQ) to capture the multiple health-relevant dimensions of employment relationships. We adopt a specific definition of EQ from Van Aerden and colleagues, which consists of seven conceptual dimensions encompassing both contractual and relational employment: (1) employment stability, (2) material rewards, (3) workers’ rights and social protections, (4) standardized working time arrangements, (5) training and employability opportunities, (6) collective organization, and (7) interpersonal power relations [15,16,17] (See Appendix A). The EQ concept originated as a subcategory within the larger study of job quality [20] but has specifically developed within public health research alongside recent multidimensional conceptualizations of precarious employment: these constructs overlap significantly in terms of the underlying theory, both emphasizing worker–employer power relations as a critical determinant of health [17]. Poor EQ and precarious employment are theorized to adversely affect worker health through several pathways, namely psychological stress (e.g., feelings of insecurity, powerlessness), exposures to hazardous physical and psychosocial working conditions (e.g., due to receiving less or worse-quality training, support, and equipment to complete work safely), and material deprivation (e.g., inadequate or unstable income and benefits) [17,18].

One important distinction within health research on precarious employment versus EQ is how these constructs have been operationalized for empirical analysis. Researchers have typically measured precarious employment using continuous measures, assuming that multiple dimensions of employment relationships can be aggregated onto a linear continuum of low to high precarity (e.g., [21]). In contrast, EQ has typically been measured using typological approaches, identifying normative classes (e.g., [15]) or trajectories (e.g., [22]) of specific configurations of employment. By using typological measurement approaches, a strength of the EQ literature is to explicitly acknowledge that employment relationships can depart from the idealized SER concept in multiple ways [16,23]. Indeed, as documented within the labor segmentation literature, beneficial and adverse EQ features tend to cluster together within different jobs, firms, and occupations, as a result of employers applying different strategies to lower costs, shield themselves from liability, and increase productivity [23], and because workers experience employment as a package of good and bad EQ elements, their health may depend on particular patterns thereof. Thus, some within the EQ and health literature view precarious employment as one particular state within a non-linear spectrum of EQ and attempt to account for the reality of heterogeneous patterns of employment [15,18].

Consistent with these rationales, the emerging literature examining EQ as a typology has provided valuable insights into linkages between employment relations, workers’ health, and social inequality. In particular, this research found substantial segmentation of employment circumstances across working populations, which varies across both contractual and relational employment features, is more nuanced than a simplistic dichotomy of standard vs. non-standard work arrangements and has some consistencies in patterning between the E.U. and U.S. labor markets [15,16,18,22,24,25,26]. The identified EQ typologies were associated with physical and mental health outcomes [15,18,22,24,25], with some studies finding these associations remain after controlling for intrinsic conditions of the work environment such as environmental or psychosocial exposures [15,24]. EQ types are also unevenly distributed across worker sociodemographic profiles, with workers from traditionally marginalized groups—including women, as well as racialized and less-educated populations—overrepresented in poor-quality forms of employment [16,22,24,26,27]. However, more research is needed to specifically investigate the role of EQ in contributing to health inequities.

### 1.2. Gender, Employment Quality, and Health

The quality and character of employment differ across gender [10,14,28]. In nearly all societies, women are selected into a limited range of occupations seen as women’s jobs (e.g., healthcare, childcare) [29]. Women also tend to occupy jobs at lower levels of the decision-making hierarchy within occupations, firms, and worksites [30]. While the roots of the gendered division of paid labor have well-established historical bases and are multifactorial, several well-described social mechanisms are interrelated. Women experience gender-based discrimination that disadvantages them with regard to hiring, wage levels, and promotional opportunities [31]. Additionally, despite participating in the labor market at similar levels to men, women today continue to shoulder a greater burden of household tasks and dependent care duties [32,33,34]. Among other impacts, the unequal distribution of unpaid labor limits women’s bargaining power and agency when seeking employment, as they may experience comparatively higher pressure to balance multiple roles associated with professional and familial domains.

As a result, women generally work in jobs with lower EQ compared to men. There is extensive evidence that occupations in which women are overrepresented are paid less, even after adjusting for required skills or levels of education [35]. Women are also decidedly more likely to work in jobs with temporary contracts and part-time hours [10,14], which are less likely to have fringe benefits and prospects for advancement. Other studies have found that women are less likely to have access to learning opportunities [36], control over their work schedules [37], be involved in workplace decision-making [14], and face higher levels of workplace harassment [38]—all of which can be indications of poor worker–employer relations.

There is also some indication that EQ produces differential health effects between women and men. Asymmetrical social expectations related to household responsibilities and caregiving may make it more difficult for women to balance work and family spheres [9,11,39]. For instance, women often take on a ‘second shift’ of unpaid labor in addition to their formal employment [40], which could adversely affect both physical and mental health [9,32]. This double duty can also constrain women’s choices and ability to engage in healthy behaviors, such as exercise, diet, and sleep [41,42]. Several studies have found that women in various forms of non-standard employment arrangements (e.g., temporary, part-time) report higher levels of poor general and mental health compared to men [43,44]. Likewise, Vives et al. [45] found women suffer worse mental health when examining a multidimensional measure of precarious employment. On the other hand, such arrangements may provide additional flexibility (from the worker’s perspective) to accommodate obligations from both work and family domains. Therefore, a health advantage might be observed in relation to some non-standard employment types for some women. Yet, scholars have noted that work-related flexibility is a gendered phenomenon. Women engaged in more flexible, non-standard arrangements may experience conflict in both work and family domains, being unable to fulfill the roles of the ideal mother or ideal worker, for example [46]. Further highlighting the complex gender dynamics of EQ and health, there is also some evidence suggesting that cultural expectations may produce comparatively worse health among men engaged in non-standard forms of employment. For instance, men may be more likely to consider stable employment as their normative societal role and experience stigmatization or threats to their masculinity when engaged in jobs that deviate from full-time, permanent employment [47].

### 1.3. The Current Study

In summary, EQ is increasingly recognized as an important determinant of health that may contribute to gender inequities in health due to a disproportionate burden of poor-quality employment experienced by women. However, little research has specifically investigated the potential role of EQ in contributing to such inequities. Further, prior research has identified the value of operationalizing EQ with typological and multidimensional measurement approaches to better account for the substantial heterogeneity in the configurations of contemporary employment relationships and to be more theoretically grounded in how workers experience employment (i.e., as a package of simultaneously occurring features). However, typological variables are often difficult to incorporate into commonly applied analytic methods to investigate health disparities, especially mediation analyses. Thus, further methodological development is needed to accommodate typological measures in such analyses.

In this study, we extend the existing EQ and health literature by explicitly examining whether EQ contributes to gender health inequities while accommodating a typological operationalization of EQ. To accomplish this, we conducted a causal mediation analysis in which we hypothesize EQ as a mechanism of the relationship between gender and two health indicators (self-reported health and frequent mental distress) among a representative sample of U.S. wage earners in the General Social Survey (2002–2018). Among its benefits, the causal (or counterfactually-based) mediation framework can accommodate nonlinear relationships, allowing us to include a latent class analysis (LCA)-derived measure of EQ (i.e., a nominal variable) as a mediator. Further, this approach allows for the estimation of indirect effects in the presence of exposure–mediator interaction, which may be important, as described above, if EQ affects women and men differently.

Figure 1 presents the directed acyclic graph (DAG) guiding this study. The total effect of gender on general and mental health outcomes is decomposed into a pure direct effect and a total indirect effect through EQ, adjusting for confounders and allowing for EQ–gender interaction. To better understand the contribution of interaction to mediating effects, the total indirect effect is further decomposed into the pure indirect effect and mediated interaction effect. We note that several other important factors lie on the causal pathway between gender and health that are not included in Figure 1, for example, education, industry, socioeconomic status, or household composition. We conceptualize such factors as being subsumed in direct and indirect effect estimates in our analysis, focusing in this study on the plausibility of EQ as a meaningful mechanism of the gender–health relationship and a potential focus of intervention to reduce gender inequities in health. Furthermore, our DAG was simplified to include only measured variables, but we recognize the potential for unmeasured confounding.

## 2. Methods

### 2.1. Data

This study uses five waves of data (2002, 2006, 2010, 2014, and 2018) from the General Social Survey (GSS), a nationally representative, repeated cross-sectional survey of American adults administered primarily via in-person interviews [48]. In the selected years, the GSS incorporated a module on the Quality of Work Life (QWL), which evaluates a wide assortment of employment and working conditions. Our population of interest in this study is wage earners; our initial pooled sample consisted of 6421 respondents that indicated that they (1) were currently employed and (2) did not identify as self-employed. Survey weights provided in the GSS are applied to all analyses to adjust for differential probabilities of selection and non-response.

### 2.2. Health Measures

Given the wide-ranging influences of EQ on health, we examine two broad indicators of general and mental health. Self-rated health is considered a good indicator of mortality and morbidity [49] and was previously associated with different patterns of EQ in the E.U. [15,24,25] and the U.S. [18,22]. Self-rated health (SRH) is measured by the question: “In general, would you say your health is …?”, which we dichotomized to distinguish fair/poor (“poor”) and good/very good/excellent (“good”) health. Dichotomizing SRH increases reliability and reduces measurement error, especially among respondents from marginalized sociodemographic groups [50].

Mental health was also associated with EQ [15,18,22,24,25] and may operate in different ways across gender and specific patterns of EQ, compared to the general health indicator. Frequent mental distress (FMD) is measured using the following item from the Centers for Disease Control and Prevention (CDC) health-related quality of life index (HRQOL) instrument: “Now thinking about your mental health, which includes stress, depression, and problems with emotions, for how many days during the past 30 days was your mental health not good?” FMD is included as a dichotomous measure, with presence defined as 14 or more mentally unhealthy days [51,52].

### 2.3. Construction of Employment Quality Measure

The EQ variable is constructed with LCA, a typological measurement approach that identifies unobserved (i.e., latent) subgroups within a population based on patterns of responses to a set of observed indicators related to the studied phenomena (in this case, EQ). We operationalize the EQ construct using 11 proxy indicators available in the GSS, spanning each of the seven conceptual EQ dimensions mentioned above (see Appendix A). These indicators are similar to those used in several prior investigations of EQ and health [15,24,25]. Respondents that did not have information on at least two EQ indicators were excluded (*n* = 32); over 97% of respondents had information for at least nine of eleven EQ indicators (total *n* = 6389). LCA modeling was conducted using Mplus (Version 8) [53].

As reported in detail elsewhere ([18,26]), we identified six distinct EQ types among wage earners in the U.S. based on a combination of model fit and substantive interpretation. Each EQ type is assigned a label corresponding to the character of the employer–employee relationship and is described briefly in Table 1. For additional information about the LCA procedure including fit statistics and model selection process, as well as analyses of sociodemographic and health correlates, see Peckham et al., 2019 [18] and 2022 [26].

### 2.4. Gender and Other Covariates

In the GSS, the variable ‘sex’ is recorded as male or female primarily by interviewers’ observation; we therefore argue that the concept of gender is more relevant to this form of data collection (i.e., based on appearance). Further, we posit that social processes and structural factors that create differential experiences and outcomes—related to health, employment, and other life domains—across women and men are fundamentally grounded in normative gender rules and culturally enforced standards [54]. For these reasons, we emphasize the concept of gender in this study and presume that ascertainment of binary sex in the GSS is concordant with gender among study respondents.

Several demographic variables are included as hypothesized confounders of the EQ–health relationship: race/ethnicity (non-Hispanic white, non-Hispanic African American, Hispanic, Asian/Pacific Islander, American Indian/Alaskan Native), nativity (born in the U.S., born outside of the U.S.), and age (<30 years, 30–50 years, >51 years). Historically, workers in the U.S. labor market were discriminated against based on both their race/ethnicity and nativity, thus these factors are correlated with EQ. Survey wave is included in all models to account for potential year effects.

### 2.5. Analytic Approach

To explore whether EQ plays a mediating role in the gender–health relationship, we used a counterfactual causal mediation framework. Briefly, this involves examining whether expected outcomes (poor SRH or FMD) change under conditions in which exposure (gender) and mediator (probability of membership within EQ type) variables are manipulated to represent relevant counterfactual scenarios. We proceeded in two stages. First, we present a two-way decomposition, in which the total effect of gender on health is decomposed into two components: the pure direct effect (PDE) and the total indirect effect (TIE). In this study, the PDE is the expected difference in health between women and men while the distribution of EQ membership is kept at the levels observed among men; the PDE is interpreted as the effect of gender on health not mediated by EQ. The TIE is the expected difference in health among women if they experienced EQ at levels observed in men compared to levels of EQ actually observed among women; the TIE is interpreted as the effect of gender on health mediated by EQ. Secondly, we further decomposed the TIE into two separate components based on VanderWeele’s three-way decomposition methodology, which characterizes the role of interaction in contributing to the indirect effect [55]. The two components of the TIE are the pure indirect effect (PIE) and the mediated interaction (INTmed). The PIE is estimated as the change in expected health status among men if they experienced the distribution of EQ observed in women compared to their actual observed distribution of EQ; the PIE is interpreted as the portion of the TIE due to mediation only. The INTmed is calculated as the difference between TIE and PIE, and is equivalent to the product of (1) an additive interaction between gender and EQ on the health outcome and (2) the average effect of gender on EQ; this is interpreted as the portion of the indirect effect due to interaction. We note that, while the decomposition of causal effects from socially defined and immutable characteristics has been met with some hesitancy within epidemiology [56], in this study, we interpret the indirect effect as a disparity reduction that may be possible through intervention on the EQ variable (e.g., making EQ more equitable across gender). Likewise, we interpret direct effect estimates as a disparity residual that would remain after such an intervention. As described by VanderWeele and Robinson [57], this interpretation requires weaker identifying assumptions than if causal effects are assigned to gender, rather than only stipulated for the mediating variable. We do recognize that we are not able to satisfy all assumptions required in causal mediation analysis (e.g., adjusting for all mediator—outcome confounders influenced by the exposure), resulting in some bias in our effect estimates. Regardless, we believe our analysis sheds some light on the potential health impacts of the intervention on EQ, a construct that has many opportunities for intervention.

Important for our purposes, a counterfactually-based mediation framework can accommodate modeling with a nominal mediator, which can be implemented within Mplus using mixture modeling [58,59]. Mechanically, this involves the joint estimation of two separate regressions. First, the influence of gender on EQ is modeled by multinomial logistic regression, with EQ as the dependent variable. Second, the health outcome (the dependent variable) is related to gender (reference: men) and EQ (reference: SER-like) using logistic regression. The latter is specified to allow for gender–health estimates within each level of the EQ variable (i.e., interaction). Consistent with current best practices for modeling relations between LCA-derived variables and auxiliary covariates or outcomes, we applied the three-step correction approach developed by Vermunt [60] during all regression analyses; this approach accounts for model classification error while maintaining the character of the identified latent variable [58,61].

Prior to regression analyses, 22 participants that did not provide information on age were excluded. Additionally, 32 participants did not report SRH, and 71 did not report FMD; these individuals contribute information related to the estimation of the associations between gender and EQ class membership probabilities, but not regression coefficients predicting health outcomes. Total, direct, and indirect effect estimates derived from counterfactual analyses are presented as probability differences; this has the advantage over relative (e.g., odds ratio) or proportional (e.g., proportion of total effect mediated) measures of not obscuring overall effect sizes [62]. Odds ratios for relevant measures are also provided for descriptive purposes. We used bias-corrected bootstrapping (*n* = 10,000) to construct 95% confidence intervals for all estimates, as is often recommended for examining indirect effects [63]. Evidence for a mediating effect of EQ is determined if any of the TIE, PIE, or INTmed estimates are non-zero. For further description of the mediation models, we also present model-predicted probabilities of EQ membership across gender, and gender–EQ probabilities of reporting poor health outcomes. These values can supplement information provided by the indirect effects estimates of the primary analysis, which are interpreted at the level of the entire EQ variable (i.e., what is the change in expected outcome when the distribution of all six EQ types is manipulated?) and allow for further examination of relations between gender and health within different patterns of EQ.

## 3. Results

Sample characteristics are presented in Table 2. Women and men differed in terms of race/ethnicity, and nativity status, but not by the trichotomized age variable. There are also clear differences in the distribution of EQ across gender based on modal assignment into most likely class. Based on crude prevalence, there is no gender difference in reporting poor SRH; however, women report slightly higher levels of FMD compared to men. The distribution of individual EQ indicators is presented by gender in the Appendix A. Women reported lower levels of income, overall and required extra hours worked, opportunity to develop, union representation, and schedule control, as well as higher levels of workplace harassment. Women were also slightly more likely than men to be in a regular, permanent arrangement, and no gender difference was seen in decision-making involvement (Appendix A).

The results from mediation analyses are presented in Table 3. The total effect of gender on SRH is not significant, with the women–men probability difference (PD) estimated as −0.003 (95% CI: −0.021, 0.015). However, our results are consistent with EQ having a significant mediation role in the gender–SRH relationship, as indicated by the non-zero TIE (PD = 0.015; 95% CI: 0.005, 0.028). In this case, we estimate that the probability of reporting poor SRH among women is 1.5% higher than if they experienced the same EQ as men, which corresponds to a 1.22-fold increased odds of poor SRH among women due to EQ. Further decomposition of the TIE suggests that the indirect effect is primarily due to differential exposure of women to various EQ types (PIE: PD = 0.023; 95% CI: 0.010, 0.038), rather than a presence of mediated interaction (INTmed: PD = −0.008; 95% CI: −0.024,0.008). Additionally, a negative PDE (PD = −0.018; 95% CI: −0.036, −0.001) suggests that women would report lower levels of poor SRH compared to men if the entire working population experienced the distribution of EQ observed in men. Thus, the results from the SRH model resemble what MacKinnon et al. [64] label ‘inconsistent mediation’, in which the direct and indirect effects have opposite signs. The combination of the PDE and TIE having similar magnitudes but opposite signs is consistent with a non-significant total effect.

In the mediation model examining FMD (Table 3), we found a significant total gender effect: women reported 3.9% higher FMD than men (95% CI: 0.009, 0.069). As with SRH, we find evidence of a significant indirect gender-FMD effect that operates through EQ (TIE: PD = 0.026; 95% CI: 0.006, 0.046). Our model suggests that women report FMD at a probability that is 2.6% higher than if they experienced the same distribution of EQ as men; this corresponds to a 1.21-fold higher odds of FMD among women due to EQ. However, unlike SRH, the EQ-attributed indirect effect on FMD is primarily due to mediated interaction (PD = 0.029; 95% CI: 0.004, 0.055). This suggests that gender differences in EQ–health associations that arise from an unequal distribution of EQ are driving the mediating effect of EQ on FMD.

Further detail of the mediation models can be gleaned by examining predicted probabilities for reporting poor SRH or FMD within each EQ type, as well as EQ membership probabilities, by gender (Table 4). As expected, these results show gender as a strong predictor of EQ. In particular, women are significantly less likely to be members of the Portfolio, Inflexible skilled, or Dead-end EQ categories. Additionally, the risk of poor SRH or FMD varies across EQ types. However, the mediation models find only weak evidence that individual EQ types differentially affect women’s and men’s health. Just one EQ–gender interaction was found to be statistically significant: women engaged in jobs resembling the Optimistic precarious EQ type report better SRH compared to men in the same type of employment. The lack of statistical significance across individual interaction coefficients is seemingly inconsistent with the finding of a statistically significant INTmed estimate within the FMD model. The results in Table 4, however, show that the overall distribution of EQ is configured such that women are heavily selected into EQ types in which they report higher levels of FMD (i.e., over 75% of women are engaged in either SER-like, Precarious, or Optimistic precarious types) and are underrepresented within EQ types in which they fare better. Thus, when the aggregate distribution of EQ across gender is considered, differential effects of EQ on FMD seem to be important for gender health disparities.

## 4. Discussion

In this study, we performed an exploratory mediation analysis to investigate whether different patterns of EQ contribute to gender inequities in two broad indicators of general and mental health among U.S. workers. While both genders reported similar overall levels of poor SRH, our analysis suggests that this apparent equality is concealing the fact that if the distribution of EQ among women was more like men’s, working women’s probability of reporting poor SRH would be reduced and they would report comparatively higher levels of SRH. This finding highlights the value of applying decomposition approaches even in the absence of a disparity. Our analysis of FMD also indicates an important role of EQ. The total effect estimate indicated women in our study reported higher levels of FMD compared to men, and this was mostly explained by an unequal distribution of EQ combined with differential impacts of EQ across gender. Overall, these results are consistent with EQ as a plausible proximal mechanism between gender and health, which may contribute to observed gender inequities in health among working populations and serve as a locus of intervention to reduce these inequities.

Our finding that EQ may operate in different ways between gender and the two examined health outcomes (i.e., different patterns of direct and indirect effects) might be at least partially due to the nature of our data. For one, the two health outcomes we examined may reflect different aspects of the gender–EQ–health relationships. We previously reported that specific EQ–health mechanisms were more salient to certain health consequences; for example, material deprivation and employment-related stressors better-explained associations of EQ with FMD compared to SRH or occupational injury [18]. Additionally, subjective ascertainment of both EQ indicators and health outcomes might contribute to different patterns across outcomes, as a survey participant’s mental health state may influence their responses during data collection. That is, a worker reporting poor mental health may be more likely to also perceive their employment circumstances as poor, compared to a worker in a better mental health state, which may contribute to the finding of overall stronger gender mediation effects of EQ on FMD. It is not surprising then, that once we incorporate gender into the analysis, which could potentially interact with either or both of these data-related dynamics in complex ways, and that we could see different patterns of associations across the examined outcomes. Nevertheless, further research is needed to better understand some of the more complicated aspects of our findings, such as women reporting better SRH but worse FMD within the optimistic precarious EQ type (Table 4).

Another counterintuitive finding was a statistically significant INTmed effect in our FMD model despite the lack of any significant individual gender–EQ interactions. It seems that when interpreting the indirect effects of a nominal mediator, it is possible that several non-significant interactions between the exposure and specific mediator categories aligned in the same direction can contribute to a significant INTmed. Alternatively, one can imagine a scenario in which several significant individual interaction effects are found, but are in opposite directions; in that case, the model may not identify a significant INTmed estimate at the variable-level. A potential insight based on our experience using an LCA-derived mediator variable within a causal mediation framework is the value of examining individual model probability estimates (as shown in Table 4). These values can supplement information provided by the indirect effect estimates, which are interpreted at the level of the entire nominal mediator variable (e.g., what is the change in expected outcome when the distribution of all six EQ types is manipulated?). Examining relationships between the individual categories of the nominal mediator may therefore be useful to understand the specific mediator categories which require more policy or programmatic attention. More generally, this finding points to the novelty of mediation modeling with a latent (nominal) mediator, which has rarely been approached within the epidemiologic literature and is not easily conducted with many popular statistical packages. We believe that typological measurement approaches are useful for examining complex multidimensional constructs and are eager to see further methodological development in this area.

### 4.1. Application of Mediation Analysis Framework to Examine Potential Role of EQ in Gender Health Inequities

A primary contribution of this study is the application of mediation to explicitly examine the plausibility of EQ as a proximal contributor to gender health inequities. Several prior mediation analyses have identified simple measures of employment status (e.g., whether engaged in paid employment, full vs. part-time) as important mediators explaining gender inequities in SRH—often more important than education or income [65,66,67]. However, such measures do not fully account for the multidimensional nature of worker–employer relations. To our knowledge, this is the first study to examine a more comprehensive measure of the employment relationship as a mediator between gender and health.

The results of this mediation analysis generally align with numerous studies that have focused on individual components of the gender–EQ–health pathway. Namely, substantial evidence suggests that women are disproportionately exposed to poor EQ features (e.g., [14]), while an emerging evidence base is finding that various individual or multidimensional measures of poor EQ (and similarly conceptualized constructs of insecure, flexible, and precarious employment) adversely affect workers’ health [1]. Such findings infer a mediation relationship in which EQ may differentially burden women’s health. Existing studies that have applied a gender lens to examine potential health inequities associated with poor-quality employment have often stratified by gender or included gender–employment interactions, i.e., evaluating effect modification or interaction of EQ by gender. However, gender-specific EQ–health associations have been inconsistent, with individual analyses finding either women or men are more adversely affected and others finding no gender differences [68]. More importantly, such analyses, which are focused on whether health associations differ across gender, do not explicitly consider whether EQ contributes to disparities in health across gender. Given the complex linkages between gender, employment, and health, a strength of our analysis is formally examining the plausibility of EQ as a mechanism of gender health inequities using a mediation approach that simultaneously considers both the gendered distribution of poor-quality employment and potential differences in health associations.

### 4.2. Accounting for Heterogeneity and Multidimensionality of Employment Relationships to Understand Gendered Labor Market Inequities

Another strength of our analysis is the use of a multidimensional, typological measure of EQ. This approach conceptualizes jobs as constellations of many different health-relevant employment features—including less-often studied components concerning worker–employer power relations—and allows for a more nuanced examination of gendered experiences in employment than can be accomplished by focusing on one EQ feature at a time or focusing solely on contractual aspects of employment. Using this measure, we found substantial segmentation of EQ across women and men—both in terms of contractual dimensions and power relations—with an overall patterning that is generally consistent with gender inequalities in the division of unpaid labor and labor market bargaining power.

Regarding the distribution of contractual EQ dimensions, our results show women are heavily selected into Precarious and Optimistic Precarious employment types (an estimated 47% of all women wage earners compared to 26% of men), which are characterized by having the highest probabilities of non-permanent contracts, low wages, low hours, and irregular schedules. Meanwhile, men-dominated forms of employment, including Portfolio, Inflexible skilled, and Dead-end EQ types (an estimated 54% of men compared to 24% of women), are distinguished as a group by having the highest probabilities of working long (and mandatory extra) hours and receiving high income. These findings align with pervasive social expectations and societal gender roles in which women assume a larger proportion of household work and caregiving—which, among other impacts, hinders their ability to engage in permanent, full-time, regularly scheduled employment—while men act as the primary family wage earner and are unequally rewarded for laboring long hours, reinforcing labor market inequalities [69].

We also find that employment-related power relations are unequally distributed across women and men. It is notable that women are particularly less likely to be engaged in the Portfolio EQ type (an estimated 18% of men compared to 5% of women). Besides involving excessive work hours, Portfolio employment represents the most privileged EQ type in terms of wages, decision-making authority, opportunity, schedule control, and other EQ features. The finding of very few women in this EQ type is consistent with the existence of a “glass ceiling” that acts as a barrier for women in obtaining positions of authority, power, and other prized employment-related benefits [70]. Likewise, the high proportion of working women being selected into Precarious employment (an estimated 22% of women compared to 11% of men) may be a result of gender discrimination and other sources of disadvantage related to bargaining power facing women. However, an additional observation detected by our typology is that women are also disproportionately selected into Optimistic precarious jobs (an estimated 26% of women compared to 15% of men), a highly non-standard form of employment from a contractual perspective but which offers a relatively high degree of schedule control, involvement in decisions, and opportunity to develop, as well as low levels of workplace harassment. This suggests that at least some working women in this EQ type are engaged in these jobs voluntarily, possibly in an effort to better balance work and non-work life spheres. Taken as a whole, we believe our measurement approach captures important aspects of the intricate relationships between gender, health, and specific combinations of EQ.

### 4.3. Limitations and Future Directions

It is important to emphasize that our intention with this exploratory analysis is to add to the theoretical and methodological base, and we are not making causal claims due to the cross-sectional nature of these data. Such concerns are amplified within the context of mediation analysis, where the temporal ordering of the study variables is critical to estimating and interpreting direct and indirect effects. While gender is clearly antecedent to employment and health, we further believe that there is a strong theoretical rationale for presuming current employment conditions are an important influence on future health status. There is substantial literature documenting the link between job conditions and individuals’ health; for example, several studies examining temporary or precarious employment contracts have found that the employment-to-health causal direction is likely more important than health selection effects [44,71]. However, reverse causation in which individuals with worse health are selected for worse EQ is also possible and is not accounted for in our study.

In addition to the cross-sectional design of our study, the use of gender, a nonmanipulable exposure may be troubling to some. Based on the guidance by VanderWeele and Robinson [57], our goal was to refocus attention on the highly manipulable mediator, EQ. However, we are aware that this analysis likely violates other assumptions required for causal mediation analyses, especially the absence of confounders of the mediator–outcome association that is affected by the exposure and other unmeasured confounders.

In terms of identifying future research directions, another limitation of this analysis is our inability to incorporate information on potential mechanisms that contribute to biases and unequal power across women and men, including the asymmetrical burden of unpaid labor, intersectionality with other sources of marginalization, and the societal-level regimes shaping these dynamics. In particular, occupational health research that has accounted for family roles and demands has shown that unpaid labor affects working women and men in distinct ways—typically contributing to greater stress and ill health among women [9,32,39]. Thus, future studies that include information on workers’ household circumstances such as family composition, housework, or dependent care responsibilities could further elucidate dynamics between gender, EQ, and health [72]. We also did not directly account for aspects of social class or socioeconomic status that may arise separate from EQ but which could capture individual-level sources of dis/advantage, resources, and power that we expect to be important for both exposure and susceptibility to poor-quality employment [33]. Another important consideration is the intersectional nature of gender with other axes of inequality, including race/ethnicity. For example, Andrea et al. [73] recently examined EQ across intersecting gender–racial/ethnic–educational subgroups of older workers in the U.S., finding that racialized women generally experienced the worst EQ compared to any other subgroup, including white women within similar educational strata. Lastly, recent cross-national comparison studies of EQ have emphasized the importance of societal-level contexts, finding that poor EQ and associations with health by gender tend to vary across measures of welfare regime types [74,75,76]. Given variations across national contexts, Fujishiro et al. [74] argue that the relationships between gender and inequities in EQ and health cannot be fully understood without considering a macro-level perspective which can provide an understanding of how paid and unpaid labor is performed in society—for example, through the character of expenditures to support families (e.g., public daycare) or labor market penalties for women (e.g., gender inequality in labor market participation). Future research will need to incorporate information on these contexts to more fully elucidate the role of employment in producing gender inequities in health.

Despite these limitations, the GSS is one of the richest sources of information related to the character of modern employment relationships in the U.S., including both contractual and relational features. This allowed for what we believe is the first study to examine gender health inequities related to a detailed, theoretically grounded measure of the employment relationship. Nevertheless, these findings will need to be replicated within data that allow for better causal inference and information on additional social contexts and mechanisms that produce gender inequalities in employment and health.

## 5. Conclusions

Despite longer life expectancy, women continue to report worse physical and mental health compared to men. In this study, we found that if the distribution of beneficial and adverse employment circumstances among women were more like that which men currently experience, women’s health would improve, positioning the character and quality of the employment relationship as a potential target for intervention to reduce gender inequities in health. Rather than focusing on individual employment conditions, future research at the intersection of gender, employment, and health should consider the multiple health-relevant aspects of the employment relationship, as well as the reality that these aspects cluster together in particular ways within the modern labor market.

## Figures and Tables

**Figure 1 ijerph-19-11237-f001:**
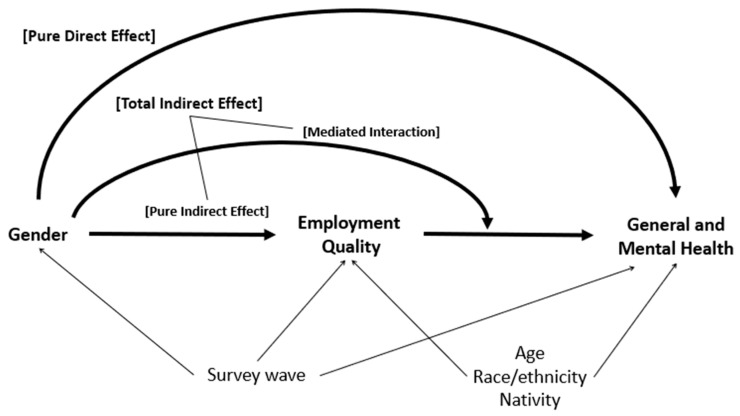
Directed acyclic graph (DAG) with additional labels to identify the direct and indirect effects. The total effect of gender on health is decomposed into a pure direct and a total indirect effect through employment quality (EQ), adjusting for confounders and accounting for gender–EQ interaction. The total indirect effect is further decomposed into the pure indirect effect and mediated interaction effect.

**Table 1 ijerph-19-11237-t001:** Employment quality (EQ) types constructed from latent class analysis.

Label	% of Wage Earners	Character of EQ Type
SER-like	28	These jobs have generally favorable EQ features, including a permanent arrangement, full-time hours, adequate wages, working during the day shift, and with adequate information/equipment to complete work. They also have moderate levels of schedule control and development opportunities and are unlikely to experience excessive work hours or workplace harassment.
Portfolio	17	These jobs have permanent arrangements, standard day shifts, and low levels of harassment; however, they are distinguished by having the highest income, employee involvement, development opportunity, and degree of schedule control of all of the EQ types. On the other hand, they work long hours.
Inflexible skilled	15	The jobs have high wages, opportunities to develop abilities, union representation, and involvement in decision-making; however, these jobs are also characterized by a high probability of irregular shifts, low schedule control, workplace harassment, long hours, and mandatory extra work.
Dead-end	14	These jobs have permanent arrangements with adequate wages and high union representation; however, workers in these jobs generally experience poor worker–employer relations. This EQ type has the lowest levels of opportunity to develop and is most likely to report inadequate information, equipment, and training to perform their work. They also lack control over their schedule or involvement in decision-making, and experience high workplace harassment.
Precarious	13	These jobs have generally poor EQ features. From a contractual perspective, these jobs have a high probability of non-permanent working arrangements, low wages, non-full-time hours, and irregular shifts. From a relational perspective, this group has very low development opportunity, schedule control, union representation, and employee involvement, as well as experiencing high workplace harassment.
Optimistic precarious	13	This type is like the *Precarious* EQ type in terms of contractual features; however, they experience high levels of development opportunity, schedule control, and employee involvement, as well as having a lower probability of encountering harassment at work.

Notes: EQ is operationalized using 11 proxy indicators spanning seven dimensions of EQ described by Van Aerden et al. (2014, 2016) [15,16]. See Appendix A for more information on dimensions, indicators, and operationalization. More information about the LCA procedure is reported in Peckham et al. (2022) [26].

**Table 2 ijerph-19-11237-t002:** Characteristics of study sample from General Social Survey: Frequency (percent).

Measure	Level	Total	Women	Men	*p*-Value ^b^
		*n* = 6367 ^a^	*n* = 3405	*n* = 2962	
Survey wave	2002	1542 (24)	803 (24)	739 (25)	0.536
	2006	1489 (23)	801 (24)	688 (23)	
	2010	1011 (16)	562 (17)	449 (15)	
	2014	1068 (17)	568 (17)	500 (17)	
	2018	1257 (20)	671 (20)	586 (20)	
Age	30 and under	1521 (24)	796 (23)	725 (24)	0.167
	31–50	3077 (48)	1630 (48)	1447 (49)	
	Over 50	1769 (28)	979 (29)	790 (27)	
Race/ethnicity	White	4339 (68)	2255 (66)	2084 (70)	<0.001
	Black	1004 (16)	624 (18)	380 (13)	
	Hispanic	747 (12)	380 (11)	367 (12)	
	Asian/Pacific Is.	202 (3)	103 (3)	99 (3)	
	AI/AN ^c^	75 (1)	43 (1)	32 (1)	
Nativity	Born in U.S.	5629 (88)	3050 (90)	2579 (87)	0.002
	Foreign born	738 (12)	355 (10)	383 (13)	
Employment quality type ^d^	SER-like	2086 (33)	1231 (36)	855 (29)	<0.001
	Portfolio	1041 (16)	378 (11)	663 (22)	
	Inflexible Skilled	847 (13)	349 (10)	498 (17)	
	Dead-end	832 (13)	406 (12)	426 (14)	
	Precarious	768 (12)	515 (15)	253 (9)	
	Optimistic Precarious	793 (12)	526 (15)	267 (9)	
Self-rated Health	Good	5438 (85)	2896 (85)	2542 (86)	0.597
	Poor	897 (14)	490 (14)	407 (14)	
	Missing ^e^	32 (1)	19 (1)	13 (0)	
Frequent mental distress	Absent	5646 (89)	2974 (87)	2672 (90)	0.002
	Present	650 (10)	388 (11)	262 (9)	
	Missing ^e^	71 (1)	43 (1)	28 (1)	

Notes: ^a^ Subsequent to latent class analysis (LCA) modeling, 22 respondents missing information on age were removed from the analysis (13 women, 9 men). ^b^ Chi square test comparing women and men. ^c^ American Indian/Alaskan Native. ^d^ Based on assignment to most likely class from LCA model. ^e^ Respondents missing outcome data do not contribute information to estimation of mediation model coefficients involving outcome.

**Table 3 ijerph-19-11237-t003:** Total, direct, and indirect effects of gender on health mediated through employment quality, based on counterfactual definitions.

	Self-Rated Health *n* = 6335				Frequent Mental Distress *n* = 6296			
	Probability Difference	(95% CI)	Odd Ratio	(95% CI)	Probability Difference	(95% CI)	Odd Ratio	(95% CI)
Total Effect	−0.003	(−0.021, 0.015)	0.97	(0.78, 1.19)	**0.039**	**(0.009, 0.069)**	1.34	(1.07, 1.67)
Pure Direct Effect	**−0.018**	**(−0.036, −0.001)**	**0.79**	**(0.63, 1.00)**	0.013	(−0.016, 0.044)	1.11	(0.88, 1.41)
Total Indirect Effect	**0.015**	**(0.005, 0.028)**	**1.22**	**(1.06, 1.41)**	**0.026**	**(0.006, 0.046)**	**1.21**	**(1.05, 1.39)**
Pure Indirect Effect	**0.023**	**(0.010, 0.038)**			−0.003	(−0.024, 0.019)		
Mediated interaction	−0.008	(−0.024, 0.008)			**0.029**	**(0.004, 0.055)**		

Notes: Men are included in the model as the reference group for the gender exposure variable; thus, positive probability differences indicate worse expected health among women, and vice versa. Models adjusted for age, race/ethnicity, nativity, and survey wave. Bolded estimates are statistically significant based on *p*-value < 0.05. The 95% confidence intervals constructed with bias-corrected bootstrapping (*n* = 10,000).

**Table 4 ijerph-19-11237-t004:** Model-predicted probabilities of EQ distribution and reporting of adverse health status by gender and EQ type.

EQ Type	Predicted EQ Distribution ^a^				Predicted Poor SRH				Predicted FMD			
	Women		Men		Women		Men		Women		Men	
SER-like	0.285	(0.22, 0.35)	0.201	(0.15, 0.26)	0.073	(0.05, 0.11)	0.076	(0.04, 0.12)	0.107	(0.06, 0.16)	0.044	(0.00, 0.11)
Portfolio	**0.046**	**(0.03, 0.08)**	**0.175**	**(0.11, 0.25)**	0.014	(0.00, 0.05)	0.030	(0.01, 0.05)	0.081	(0.01, 0.17)	0.094	(0.05, 0.16)
Inflexible skilled	**0.116**	**(0.08, 0.16)**	**0.233**	**(0.17, 0.30)**	0.073	(0.03, 0.14)	0.055	(0.03, 0.09)	0.126	(0.05, 0.23)	0.167	(0.11, 0.24)
Dead-end	**0.080**	**(0.05, 0.12)**	**0.136**	**(0.09, 0.19)**	0.111	(0.07, 0.18)	0.153	(0.11, 0.22)	0.206	(0.12, 0.31)	0.240	(0.15, 0.34)
Precarious	0.215	(0.15, 0.30)	0.106	(0.07, 0.16)	0.160	(0.11, 0.22)	0.149	(0.08, 0.24)	0.329	(0.24, 0.42)	0.219	(0.11, 0.35)
Optimistic precarious	0.258	(0.19, 0.33)	0.149	(0.11, 0.20)	**0.078**	**(0.05, 0.12)**	**0.176**	**(0.11, 0.27)**	0.170	(0.10, 0.25)	0.136	(0.05, 0.24)

Notes: Values represent model-predicted probabilities and 95% confidence intervals constructed with bias-corrected bootstrapping (*n* = 10,000); these are interpreted as the estimated proportion of wage-earning women/men within each EQ type/reporting each health outcome. Bolded estimates correspond to statistically significant gender coefficients in mediation models (i.e., gender predicting EQ membership, or EQ-gender interactions predicting health), based on *p*-value < 0.05. ^a^ Values shown are from self-rated health (SRH) mediation model, which are nearly identical to corresponding estimates in the frequent mental distress (FMD) model.

## Data Availability

The data analyzed in this study are available from NORC at the University of Chicago, https://gss.norc.org/Get-The-Data (accessed on 14 December 2019).

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
