# Peer review of "Do Different Patterns of Employment Quality Contribute to Gender Health Inequities in the U.S.? A Cross-Sectional Mediation Analysis"

_ijerph, 2022, doi:10.3390/ijerph191811237_

Round 1

Reviewer 1 Report

Dear authors, below my suggestions about your paper.

Instead of mediated interaction, more appropriate is mediated moderation.

Additionally, there is no shown value of correlation coefficient between self-rated health and mental health? It should be added to the text to emphasize that these variables are not strongly correlated. If they are at least moderately correlated, authors maybe should consider using them as two indicators of a latent variable? It should be recognized and explained in paper. Your constatation „women reporting better SRH but worse FMD within the optimistic
precarious EQ type” leads to assumptions that SRH and FMD are at least weak correlated?

You wrote about the EQ construct consisting of nine indicators which in the result of LCA modeling led to 6 EQ types, but in Table 3, when you show the total, direct, and indirect effects of gender on health mediated through employment quality there are no 6 types of EQ. If it is the global result of EQ without division for types you should use CFA to show the EQ factor are appropriately strong loaded by those nine indicators. Please explain it to me?

I am a little bit confused. In my opinion you could conducted your analysis easier. You examine whether the type of EQ moderates the relationship between gender and SRH and FMD showing that some types of EQ are related to gender in explaining SRH and FMD?

Another situation would be if you treat EQ as one continuous variable, but the EFA should be used to show that the EQ as one factor is strong enough loaded by the 9 indicators, or maybe more than one EQ  factor exists? In this situation, when EFA confirms one EQ factor the multigroup analysis in reference to gender indicates whether gender moderates the relationship between EQ and SRH or FMD, or not. In other words, whether gender and EQ explain SRH and FMD in an interactive way, and additionally you have direct effects of EQ on SRH and FMD and gender on SRH and FMD.

If you have 6 EQ types you can compare every model with one type due to the moderating role of gender between this type and SRH and FMD? Am I wrong?

In both cases, you can use a multi-group approach based on expected differences among women and men (moderating role of gender). See this https://www.youtube.com/watch?v=cTNRgVZH20I

Author Response

We would like to express our gratitude to the reviewers for their thoughtful feedback, which has strengthened the clarity, cohesiveness and quality of our manuscript. We have incorporated the feedback in our revised manuscript. Below we provide detail about how we responded to each comment in the manuscript in bold text.

Dear authors, below my suggestions about your paper.

Instead of mediated interaction, more appropriate is mediated moderation. 

Thank you for the chance to clarify. The term “mediated interaction” comes directly from VanderWeele's three-way decomposition methodology (VanderWeele 2013), which is a specific decomposition approach within the causal (or counterfactual-based) mediation analysis framework that we applied in our study. As described in the first paragraph of Section 2.5, in the three-way decomposition approach the total indirect effect is further decomposed into two components: the pure indirect effect and mediated interaction. Mediated interaction therefore has a specific meaning, which, in brief, is interpreted as the portion of the indirect effect due to interaction. 

REFERENCES:

  1. J. VanderWeele, “A Three-way Decomposition of a Total Effect into Direct, Indirect, and Interactive Effects,” Source Epidemiol., vol. 24, no. 2, pp. 224–232, 2013, doi: 10.1097/EDE.0b013e318281.

Additionally, there is no shown value of correlation coefficient between self-rated health and mental health? It should be added to the text to emphasize that these variables are not strongly correlated. If they are at least moderately correlated, authors maybe should consider using them as two indicators of a latent variable? It should be recognized and explained in paper. Your constatation “women reporting better SRH but worse FMD within the optimistic precarious EQ type” leads to assumptions that SRH and FMD are at least weak correlated?

Thank you for this suggestion. However, it is common practice in epidemiology to individually examine associations of exposures of interest with multiple health outcomes. Indeed, we feel it is useful to have two contrasting health indicators, each of which may reflect different aspects of the gender-EQ-health relationship. And because our data are self-reported and cross-sectional, examining multiple health indicators may hedge against identifying completely spurious associations. 

SRH in particular is a commonly-examined and well-recognized health outcome within epidemiology, whereas a latent health variable may be harder to interpret for many readers. By using consistent definitions of SRH and FMD, our results can be more directly compared to other studies. We are unclear as to the value of having our outcome be a latent variable based on these two indicators. Perhaps this is a difference between scientific disciplines, but using a latent variable outcome is not common in our field, which we hope the reviewer understands.

You wrote about the EQ construct consisting of nine indicators which in the result of LCA modeling led to 6 EQ types, but in Table 3, when you show the total, direct, and indirect effects of gender on health mediated through employment quality there are no 6 types of EQ. If it is the global result of EQ without division for types you should use CFA to show the EQ factor are appropriately strong loaded by those nine indicators. Please explain it to me?

Thank you for the opportunity to clarify our approach. First, a small clarification: our EQ measure is based on eleven indicators (not nine). But, more importantly, as described in Section 2.5, the indirect effects estimates in our mediation analysis are interpreted at the level of the entire EQ variable, i.e., our model asks the question ‘what is the change in expected outcome between women and men when the distribution of all six EQ types is manipulated?’ In other words, we are indeed using a nominal variable as a mediator, which consists of six categories representing different patterns of EQ, but our mediation effects are interpreted as those related to the entire EQ variable (all six types). These are the mechanics of the counterfactual approach we’ve applied, and we believe such an analysis is appropriate for our primary objective of examining the plausibility of EQ as a meaningful mechanism of the gender-health relationship and potential focus of intervention to reduce gender inequities in health. 

To supplement the global results, we also present model-predicted probabilities of EQ membership across gender, and probabilities of reporting poor health outcomes by gender and EQ type (see Table 4). In our view, this provides additional information for the reader to further examine relations between gender and health within different patterns of EQ--beyond the variable-level indirect effect estimates.

I am a little bit confused. In my opinion you could conducted your analysis easier. You examine whether the type of EQ moderates the relationship between gender and SRH and FMD showing that some types of EQ are related to gender in explaining SRH and FMD?

Another situation would be if you treat EQ as one continuous variable, but the EFA should be used to show that the EQ as one factor is strong enough loaded by the 9 indicators, or maybe more than one EQ factor exists? In this situation, when EFA confirms one EQ factor the multigroup analysis in reference to gender indicates whether gender moderates the relationship between EQ and SRH or FMD, or not. In other words, whether gender and EQ explain SRH and FMD in an interactive way, and additionally you have direct effects of EQ on SRH and FMD and gender on SRH and FMD.

If you have 6 EQ types you can compare every model with one type due to the moderating role of gender between this type and SRH and FMD? Am I wrong?

In both cases, you can use a multi-group approach based on expected differences among women and men (moderating role of gender). See this https://www.youtube.com/watch?v=cTNRgVZH20I

From this line of questioning and suggestions, we do detect some potential points of confusion. 

First, our emphasis is on mediation, not moderation. Our goal is to evaluate EQ as a potential mechanism of observed gender-health disparities. EQ is readily modifiable (e.g., by employers or policymakers), and thus mediation analysis could help identify EQ as a potential locus of intervention to reduce such disparities. However, as discussed in our introduction, it is highly likely that the EQ-health relationship could vary between women and men, so our mediation approach will need to be able to account for the potential for interaction between gender and EQ (i.e., the exposure and mediator, respectively, in this analysis).

A second major issue is that we specifically want to operationalize EQ using a typological measurement approach. This is based on prior studies (including ours) which have identified the value in examining typologies of EQ, rather than continuous measures, which accounts for the heterogeneity of modern employment and the argument that workers’ health may depend more on specific patterns of multiple EQ features, rather than just levels/magnitudes. So our mediation approach must also be able to account for a nominal variable mediator. We quickly note that our research group has also examined EQ (and the closely related concept of precarious employment) using several other operationalizations, including continuous measures (Andre et al., 2021a, 2021b, Patil et al., 2020, Oddo et al., 2021).

So we have chosen to use a causal (or counterfactually-based) mediation approach which can accommodate both of the above stated requirements, among other advantages over traditional mediation analysis approaches. That is, this approach allows for estimation of indirect effects in the presence of exposure-mediator interaction and can accommodate nonlinear relationships, including nominal mediators. Importantly, it is our understanding that traditional mediation analyses based on path analysis cannot easily accommodate nominal mediators. However, even if there are other possible approaches, we believe that our approach is certainly appropriate for our study objective.

We have added a paragraph and additional sentences in the introduction Section 1.3 to be more explicit regarding the motivations and rationales for our chosen methodological approach.

“In summary, EQ is increasingly recognized as an important determinant of health that may contribute to gender inequities in health due to a disproportionate burden of poor-quality employment experienced by women. However, little research has specifically investigated the potential role of EQ in contributing to such inequities. Further, prior research has identified the value of operationalizing EQ with typological and multidimensional measurement approaches to better account for the substantial heterogeneity in the configurations of contemporary employment relationships and to more theoretically-ground how workers experience employment (i.e., as a package of simultaneously occurring features). However, typological variables are often difficult to incorporate into commonly applied analytic methods to investigate health disparities, especially mediation analyses. Thus, further methodological development is needed to accommodate typological measures into such analyses.

In this study, we extend the existing EQ and health literature by explicitly examining whether EQ contributes to gender health inequities, while accommodating a typological operationalization of EQ. To accomplish this, we conducted a causal mediation analysis in which we hypothesized EQ as a mechanism of the relationship between gender and two health indicators (self-reported health and frequent mental distress) among a representative sample of U.S. wage earners in the General Social Survey (2002-2018). Among its benefits, the casual (or counterfactually-based) mediation framework can accommodate nonlinear relationships, allowing us to include a latent class analysis-derived measure of EQ (i.e., a nominal variable) as a mediator. Further, this approach allows for estimation of indirect effects in the presence of exposure-mediator interaction, which may be important, as described above, if EQ affects women and men differently.” 

REFERENCES:

Andrea, S.B., Eisenberg-Guyot, J., Peckham, T., Oddo, V.M. and Hajat, A., 2021a. Intersectional trends in employment quality in older adults in the United States. SSM-Population Health, 15.

Andrea, S.B., Eisenberg-Guyot, J., Oddo, V.M., Peckham, T., Jacoby, D. and Hajat, A., 2021b. Beyond Hours Worked and Dollars Earned: Multidimensional EQ, Retirement Trajectories and Health in Later Life. Work, Aging and Retirement, 8(1), pp.51-73.

Oddo, V.M., Zhuang, C.C., Andrea, S.B., Eisenberg-Guyot, J., Peckham, T., Jacoby, D. and Hajat, A., 2021. Changes in precarious employment in the United States: A longitudinal analysis. Scandinavian journal of work, environment & health, 47(3), pp.171-180.

Patil, D., Enquobahrie, D.A., Peckham, T., Seixas, N. and Hajat, A., 2020. Retrospective cohort study of the association between maternal employment precarity and infant low birth weight in women in the USA. BMJ Open, 10(1).

Reviewer 2 Report

Dear Authors,

I appreciate the work you put into preparing the article. You have made a creative interpretation of the General Social Survey data and attempted a causal explanation of respondents' health assessments. However, I have reservations about several elements of your work, as I describe below.  

The survey is based on data collected systematically from a large group of respondents. This is an undoubted advantage. However, it entailed redefining the variables.  Dichotomization reduces measurement error but also reduces accuracy. 

The statement that "Based on a combination of model fit and substantive interpretation, we identified six distinct EQ types among wage earners in the U.S. Each EQ type is assigned a label corresponding to the character of the employer-employee relationship" is insufficient for me to accept the operationalization of the variable as valid.

The theoretical introduction gives sufficient justification for the research problem. 

However, a clear statement of the research problem is missing.

The hypotheses indicated in the text, "we hypothesize that social processes and structural factors that create differential experiences and out-comes related to health, employment, and other life domains across women and men are fundamentally grounded in normative gender rules and culturally-enforced standards [54]." is not verified. What social processes are studied? What other life domains culturally-enforced standards? Especially since the male and female samples differ in terms of the number of respondents coming from different groups of Race/Ethnicity (see Tab.2.).

Similarly, the statement that "Poor EQ and precarious employment are hypothesized to adversely affect worker health through several pathways, namely psychological stress (e.g., feelings of insecurity, powerlessness), exposures to hazardous physical and psychosocial working conditions (e.g., due to receiving less or worse-quality training, support, and equipment to complete work safely), and material deprivation (e.g., inadequate or unstable income and benefits) [17], [18]." is also untested because the study did not measure the variables listed here.

The data in Table 2 indicate that gender does not differentiate SRH ratings. So how do the authors want to explain this through mediation? 

What I find missing from the description of the results is what percentage of the variance was explained. the effect sizes of a few percent suggest the explanation of a very small portion of the variance by the variables used in the model. This lowers the Overall Merit score.

In the discussion section, its first part, there are no references to other studies and it can hardly be called a discussion, rather a summary of the results collected. There is only one reference there (line 422). Lines 415 - 423 content-wise are more suited to the section on limitations of the study, as it points out the lack of precision in the measurement of variables and confounding variables not controlled for in the study.

Author Response

We would like to express our gratitude to the reviewers for their thoughtful feedback, which has strengthened the clarity, cohesiveness and quality of our manuscript. We have incorporated the feedback in our revised manuscript. Below we provide detail about how we responded to each comment in the manuscript in bold text.

Dear Authors,

I appreciate the work you put into preparing the article. You have made a creative interpretation of the General Social Survey data and attempted a causal explanation of respondents' health assessments. However, I have reservations about several elements of your work, as I describe below. 

The survey is based on data collected systematically from a large group of respondents. This is an undoubted advantage. However, it entailed redefining the variables. Dichotomization reduces measurement error but also reduces accuracy. 

Thank you for this reminder. We presume that the reviewer is primarily concerned in their comment with our dichotomization of the health outcomes, in particular, self-reported health. In our reading of the literature, dichotomizing SRH is both extremely common and provides some analytic advantages. For example, prior research has shown that reliability of the SRH measure is worse for disadvantaged sociodemographic groups, and some scholars recommend dichotomizing as a useful strategy to improve reliability (Zajacova and Dowd 2011).

We also re-parameterized survey items to define our proxy indicators of EQ (as shown in Supplemental Table 1). Generally, our approach was to define variables based on a combination of a) using parameterizations from prior research using the same approach, b) our theoretical understanding of how EQ operates in the US (e.g., working over 48 hours per week is considered long working hours), and c) practical considerations (e.g., combining responses of ‘not too true’ or ‘not at all true’ to a response category of ‘no’). Collapsing the EQ indicators was necessary to make the LCA modeling process more manageable, and is common practice--especially with a high number of indicators.

REFERENCES:

  1. Zajacova and J. B. Dowd, “Reliability of Self-rated Health in US Adults,” Am. J. Epidemiol., vol. 174, no. 8, pp. 977–983, 2011, doi: 10.1093/aje/kwr204.

The statement that "Based on a combination of model fit and substantive interpretation, we identified six distinct EQ types among wage earners in the U.S. Each EQ type is assigned a label corresponding to the character of the employer-employee relationship" is insufficient for me to accept the operationalization of the variable as valid.

Thanks for the chance to clarify. We have recently published two other peer-reviewed articles using this typological operationalization of EQ (Peckham et al. 2019 [ref #18] and 2022 [ref #26]), which provide more information on fit statistics, model selection, and other details of the LCA modeling procedure. In the interest of space and not detracting from the focus of this paper, we think it makes sense to point readers to these articles. 

To briefly describe our approach, we first examined Akaike’s information criterion (AIC), Bayesian information criterion (BIC), Vuong-Lo-Mendell-Rubin likelihood ratio test (VLMR-LRT) to identify best fitting and most parsimonious models; in this case, fit statistics suggested around 5 classes. We then conducted a thorough substantive interpretation for three through eight class solutions for concordance with theory and prior research. This substantive interpretation is both common practice and vital to determine the most stable and meaningful model (Masyn 2013). Further, our approach follows very closely the work of numerous recently published EQ studies, in terms of the EQ conceptual framework and proxy indicators, and LCA modeling procedure--and with similar and complementary results (Van Aerden et al. 2014, 2016, 2017; Gevaert et al. 2022).

To make it more clear to the reader that we are pointing them to other publications for further detail on the LCA-based operationalization of EQ, we have modified the second paragraph in Section 2.3. Construction of employment quality measure to the following:

“As we have reported in detail elsewhere ([18], [26]), we identified six distinct EQ types among wage earners in the U.S. based on a combination of model fit and substantive interpretation. Each EQ type is assigned a label corresponding to the character of the employer-employee relationship and is described briefly in Table 1. For additional information about the LCA procedure including fit statistics and model selection process, as well as analyses of sociodemographic and health correlates, see Peckham et al., 2019 and 2022 [18], [26].”

REFERENCES:

  1. Peckham, K. Fujishiro, A. Hajat, B. P. Flaherty, and N. Seixas, “Evaluating Employment Quality as a Determinant of Health in a Changing Labor Market,” RSF Russell Sage Found. J. Soc. Sci., vol. 5, no. 4, p. 258, Sep. 2019, doi: 10.7758/rsf.2019.5.4.09.
  2. Peckham, B. Flaherty, A. Hajat, K. Fujishiro, D. Jacoby, and N. Seixas, “What Does Non-standard Employment Look Like in the United States? An Empirical Typology of Employment Quality,” Soc. Indic. Res., 2022, doi: 10.1007/s11205-022-02907-8.

Masyn, K. E. (2013). 25 latent class analysis and finite mixture modeling. The Oxford Handbook of Quantitative Methods,2, 551–611.

  1. Van Aerden, V. Puig-Barrachina, K. Bosmans, and C. Vanroelen, “How does employment quality relate to health and job satisfaction in Europe? A typological approach,” Soc. Sci. Med., vol. 158, pp. 132–140, 2016, doi: 10.1016/j.socscimed.2016.04.017.
  2. Van Aerden, G. Moors, K. Levecque, and C. Vanroelen, “Measuring Employment Arrangements in the European Labour Force: A Typological Approach,” Soc. Indic. Res., vol. 116, no. 3, pp. 771–791, 2014, doi: 10.1007/s11205-013-0312-0.
  3. Van Aerden, S. Gadeyne, and C. Vanroelen, “Is any job better than no job at all? Studying the relations between employment types, unemployment and subjective health in Belgium.,” Arch. Public Heal., vol. 75, no. 55, 2017, doi: 10.1186/s13690-017-0225-5.
  4. Gevaert, K. Van Aerden, D. De Moortel, and C. Vanroelen, “Employment quality as a health determinant: Emprical Evidence for the Waged and Self-Employed,” Work Occup., 2020, doi: 10.1177/0730888420946436.

The theoretical introduction gives sufficient justification for the research problem. However, a clear statement of the research problem is missing.

Thank you for this suggestion. We have tried to make our motivation more explicit in the introduction. We have added a paragraph and additional sentences in Section 1.3 Current study, which now reads as follows:

“In summary, EQ is increasingly recognized as an important determinant of health that may contribute to gender inequities in health due to a disproportionate burden of poor-quality employment experienced by women. However, little research has specifically investigated the potential role of EQ in contributing to such inequities. Further, prior research has identified the value of operationalizing EQ with typological and multidimensional measurement approaches to better account for the substantial heterogeneity in the configurations of contemporary employment relationships and to more theoretically-ground how workers experience employment (i.e., as a package of simultaneously occurring features). However, typological variables are often difficult to incorporate into commonly applied analytic methods to investigate health disparities, especially mediation analyses. Thus, further methodological development is needed to accommodate typological measures into such analyses.

In this study, we extend the existing EQ and health literature by explicitly examining whether EQ contributes to gender health inequities, while accommodating a typological operationalization of EQ. To accomplish this, we conducted a causal mediation analysis in which we hypothesize EQ as a mechanism of the relationship between gender and two health indicators (self-reported health and frequent mental distress) among a representative sample of U.S. wage earners in the General Social Survey (2002-2018). Among its benefits, the casual (or counterfactually-based) mediation framework can accommodate nonlinear relationships, allowing us to include a latent class analysis-derived measure of EQ (i.e., a nominal variable) as a mediator. Further, this approach allows for estimation of indirect effects in the presence of exposure-mediator interaction, which may be important, as described above, if EQ effects women and men differently.”

The hypotheses indicated in the text, "we hypothesize that social processes and structural factors that create differential experiences and out-comes related to health, employment, and other life domains across women and men are fundamentally grounded in normative gender rules and culturally-enforced standards [54]." is not verified. What social processes are studied? What other life domains culturally-enforced standards? Especially since the male and female samples differ in terms of the number of respondents coming from different groups of Race/Ethnicity (see Tab.2.).

Thank you for the chance to make this more clear. Perhaps “hypothesize” is a confusing word to use here. This statement is meant to describe our theoretical rationale for using the concept of gender, and operationalizing gender using the sex variable available in the GSS data. This statement is not about what we are testing in our analysis. We have switched the wording to “posit”, so that this sentence now reads as:

“Further, we posit that social processes and structural factors that create differential experiences and outcomes—related to health, employment, and other life domains—across women and men are fundamentally grounded in normative gender rules and culturally-enforced standards [54].”

Similarly, the statement that "Poor EQ and precarious employment are hypothesized to adversely affect worker health through several pathways, namely psychological stress (e.g., feelings of insecurity, powerlessness), exposures to hazardous physical and psychosocial working conditions (e.g., due to receiving less or worse-quality training, support, and equipment to complete work safely), and material deprivation (e.g., inadequate or unstable income and benefits) [17], [18]." is also untested because the study did not measure the variables listed here.

Again, our use of the term “hypothesized” may have caused confusion. We have changed the word to “theorized”, so that this sentence now reads as:

“Poor EQ and precarious employment are theorized to adversely affect worker health through several pathways, namely psychological stress (e.g., feelings of insecurity, powerlessness), exposures to hazardous physical and psychosocial working conditions (e.g., due to receiving less or worse-quality training, support, and equipment to complete work safely), and material deprivation (e.g., inadequate or unstable income and benefits) [17], [18].” 

Note that we have tested these pathways as potential mechanisms of the EQ-health relationship in a prior publication that used the same typological operationalization of EQ used in this study (finding that each were plausible). See Peckham et al., 2019 (reference 18). 

REFERENCES:

  1. Peckham, K. Fujishiro, A. Hajat, B. P. Flaherty, and N. Seixas, “Evaluating Employment Quality as a Determinant of Health in a Changing Labor Market,” RSF Russell Sage Found. J. Soc. Sci., vol. 5, no. 4, p. 258, Sep. 2019, doi: 10.7758/rsf.2019.5.4.09.

The data in Table 2 indicate that gender does not differentiate SRH ratings. So how do the authors want to explain this through mediation? 

Despite finding no total effect of gender on SRH, we found using our causal decomposition approach that there was a significant indirect effect of gender on SRH that operated through EQ. As we mention in the Results section, this scenario is possible when the direct and indirect effects have similar magnitudes and opposite signs. This situation is both mathematically possible and well documented to occur in the literature, and has been labeled as ‘inconsistent mediation’ or a ‘suppression effect’ (Mackinnon 2000). We believe our results therefore highlight the importance of decomposition approaches even in the apparent absence of a disparity, something others have also pointed out (e.g., Bauer and Scheim 2019).

We have added the following sentence to the first paragraph of the discussion section:

“This finding highlights the value of applying decomposition approaches even in the absence of a disparity.” 

REFERENCES:

MacKinnon, D. P., Krull, J. L., & Lockwood, C. M. (2000). Equivalence of the mediation, confounding and suppression effect. Prevention science, 1(4), 173-181.

Bauer, G. R., & Scheim, A. I. (2019). Methods for analytic intercategorical intersectionality in quantitative research: discrimination as a mediator of health inequalities. Social Science & Medicine, 226, 236-245.

What I find missing from the description of the results is what percentage of the variance was explained. the effect sizes of a few percent suggest the explanation of a very small portion of the variance by the variables used in the model. This lowers the Overall Merit score.

We don’t believe that the mediated effects in our study are small, but there are also limitations to reporting measures of mediation effect sizes. We will explain.

Rather than variance explained, a more common and somewhat analogous way to measure the relative size of a mediated effect within the causal mediation framework is with a ‘proportion mediated’ measure. (A recent review identified proportion mediated as the most commonly used effect size measure in recent publications applying mediation [Rijnhart et al. 2021].) Proportion mediated measures represent the size of the indirect effect estimate relative to the total effect estimate. For example, in our FMD model, our indirect effect was a probability difference of 2.6 percentage points and our total effect was 3.9 percentage points. The proportion of the Gender-FMD effect mediated by EQ would therefore be 2.6/3.9 = 66%, which we think is a rather large value. 

However, if all component effects are not in the same direction, as is the case in our SRH model, calculating proportion mediated measures can result in non-intuitive proportions that are negative or in excess of 100%. That said, it has been proposed that one could calculate proportion mediated in situations of inconsistent mediation by summing the absolute values of the direct and indirect effects, and dividing the absolute value of the indirect effect by this sum (Alwin and Hauser 1975). Using this approach the proportion mediated calculation for our SRH model would be |1.5|/(|1.5| + |-1.8|) = 45%. Again, this is not a small value.

In addition to the complication of inconsistent mediation in our SRH model, we are hesitant to include the proportion mediated measures in the paper because we feel such measures may be susceptible to overinterpretation. This concern is compounded by other limitations of our analyses, namely our use of cross-sectional data and the high likelihood of unmeasured confounding. For example, we think it is unlikely that two thirds of the FMD disparity between women and men is due solely to EQ, and we would not want readers to take such a measure at face value. We believe presenting these data on the absolute scale has the advantage of not obscuring the overall effect size. Ultimately, we are principally concerned with examining the plausibility of EQ as a meaningful mechanism of the gender-health relationship and potential focus of intervention to reduce gender inequities in health, which we believe this analysis does in an appropriate way.

REFERENCES:

Rijnhart, J.J., Lamp, S.J., Valente, M.J., MacKinnon, D.P., Twisk, J.W. and Heymans, M.W., 2021. Mediation analysis methods used in observational research: a scoping review and recommendations. BMC medical research methodology, 21(1), pp.1-17.

Alwin, Duane F., and Robert M. Hauser. "The decomposition of effects in path analysis." American sociological review (1975): 37-47.

In the discussion section, its first part, there are no references to other studies and it can hardly be called a discussion, rather a summary of the results collected. There is only one reference there (line 422). Lines 415 - 423 content-wise are more suited to the section on limitations of the study, as it points out the lack of precision in the measurement of variables and confounding variables not controlled for in the study.

Thank you for the chance to respond. Our intention in how we structured the discussion section was to--following a brief summary of the overall analysis, which is a common practice in our field of epidemiology--first address some of the counterintuitive aspects of our analysis and results. We thought this would be useful given the novelty and complexity of this analysis. In particular, we are concerned that readers may have lingering questions from our results--especially, concerning different patterns of findings in our two health indicators and the lack of evidence of gender-EQ interaction--and want to address these issues early. Such a strategy is not uncommon in discussion sections within the field of epidemiology. We do acknowledge that to address these topics, some of the text does end up sounding a bit like limitations. However, given our above stated concerns, we feel these paragraphs are well-placed, and that our limitations section is likewise sufficient. Further, the subsequent section (section 4.1) is specifically dedicated to contextualizing our study within the literature, including citing relevant studies. Overall, we feel that all of the aspects of a proper discussion section are included.

Round 2

Reviewer 2 Report

Thank you for your response.